# The Casual Association Inference for the Chain of Falls Risk Factors-Falls-Falls Outcomes: A Mendelian Randomization Study

**DOI:** 10.3390/healthcare11131889

**Published:** 2023-06-29

**Authors:** Jia-Xin Wu, Fei-Yan Deng, Shu-Feng Lei

**Affiliations:** 1Center for Genetic Epidemiology and Genomics, School of Public Health, Medical College of Soochow University, Suzhou 215123, Chinafdeng@suda.edu.cn (F.-Y.D.); 2Collaborative Innovation Center of Bone and Immunology between Sihong Hospital and Soochow University, Suzhou 215123, China; 3Jiangsu Key Laboratory of Preventive and Translational Medicine for Geriatric Diseases, Soochow University, Suzhou 215123, China; 4Changzhou Geriatric Hospital, Soochow University, Changzhou 213000, China

**Keywords:** causal relationship, falls, Mendelian randomization

## Abstract

Previous associations have been observed not only between risk factors and falls but also between falls and their clinical outcomes based on some cross-sectional designs, but their causal associations were still largely unclear. We performed Mendelian randomization (MR), multivariate Mendelian randomization (MVMR), and mediation analyses to explore the effects of falls. Our study data are mainly based on White European individuals (40–69 years) downloaded from the UK Biobank. MR analyses showed that osteoporosis (*p* = 0.006), BMI (*p* = 0.003), sleeplessness (*p* < 0.001), rheumatoid arthritis (*p* = 0.001), waist circumference (*p* < 0.001), and hip circumference (*p* < 0.001) have causal effects on falls. In addition, for every one standard deviation increase in fall risk, the risk of fracture increased by 1.148 (*p* < 0.001), the risk of stroke increased by 2.908 (*p* = 0.003), and a 1.016-fold risk increase in epilepsy (*p* = 0.009). The MVMR found that sleeplessness is an important risk factor for falls. Finally, our mediation analyses estimated the mediation effects of falls on the hip circumference and fracture (*p* < 0.001), waist circumference and epilepsy (*p* < 0.001), and sleeplessness and fracture (*p* = 0.005). Our study inferred the causal effects between risk factors and falls, falls, and outcomes, and also constructed three causal chains from risk factors → falls → falls outcomes.

## 1. Introduction

Falls are the most common cause of injury among the elderly. Accidental falls impose significant morbidity, mortality, and socioeconomic burden, rendering them the second leading cause of hospitalization for all age groups around the world [1]. Falls and fall-related injuries contribute to 10–15% of all emergency visits and 646,000 deaths worldwide every year. Moreover, the incidence of falls and fall-related injuries increases with advancing age and degree of frailty. Therefore, falls are common among the elderly, which increase anxiety and reduce the quality of life in the elderly [2]. Falls can also lead to serious injury, including broken bones, stroke, head injury, severe post-traumatic stress, and even death [3,4,5,6]. Hip fractures are one of the most disabling consequences of falls. According to the survey, falls contribute to more than 90% of the 250,000 hip fractures each year in the United States [7]. However, most of these observed associations were based on cross-sectional observational design, but their causal associations have not been inferred.

On the other hand, a long list of risk factors can lead to falls. For example, some neurological diseases such as Parkinson’s and Alzheimer’s disease increase the risk of falls through several mechanisms, including increased stiffness of lower body muscle tissue, motor slowing, orthostatic deficits, and in some cases, cognitive impairment [8,9]. Diabetes, cardiovascular disease, systemic comorbidities, and visual impairment were associated with falls [10,11]. An epidemiological study has shown that the risk of falls is related to waist circumference, hip circumference, and BMI [12]. A meta-analysis also showed that personality disorders are common in older adults with falls [13]. In addition, several studies have shown that people with rheumatoid arthritis, insomnia, Alzheimer’s disease, and osteoporosis are more likely to fall [14,15,16]. The associations between risk factors and falls have been observed, but the causal associations have not been inferred yet.

Mendelian randomization (MR) is an effective method for inferring causal relationships [17]. In brief, MR assumes that if exposure is causally related to outcome, genetic variants associated with exposure will also be associated with outcome via the exposure pathway. MR uses genetic variation as an instrumental variable, which is usually less confounding than in traditional observational studies, where genetic variation is randomly assigned at the time of conception [18]. This randomization process is similar to the profile in a randomized controlled trial, where participants are randomly categorized and differentially exposed. Significant differences in outcomes between groups provide evidence for a putative causal effect of exposure on outcomes. In this sense, MR is at the interface of experimental and observational studies and is referred to as the natural randomized trial [19] to generate evidence supporting the potential causal effect of exposure.

To infer causal relationships, this study performed systemic causal inference for two pairs (risk factors and falls, falls and outcomes) and further constructed the causal chains from risk factors → falls → falls outcomes using the publicly available GWAS data. 

## 2. Materials and Methods

### 2.1. GWAS Summary Statistics

According to the literature review, we selected hip circumference, waist circumference, BMI, rheumatoid arthritis, sleeplessness, type 2 diabetes, cataract, Alzheimer’s disease, depression, atherosclerotic heart disease, glaucoma, stroke, and bipolar disorder as potential risk factors for falls [8,9,10,11,12,14,15,20,21]. In addition, we chose epilepsy, fracture, headache, death, anxiety disorder, severe stress, and stroke as potential fall outcomes [4,5,6]. We downloaded the relative GWAS summary data: GWAS summary statistics of waist circumference (*n* = 336,639), hip circumference (*n* = 336,601), Waist-hip ratio (*n* = 85978), Sitting height (*n* = 336172), and osteoporosis (*n* = 337,159) are available online. (https://gwas.mrcieu.ac.uk/ (accessed on 20 June 2022)) Data on height (*n* = 253288) and weight (*n* = 336,227) were obtained from GIANT consortium data files [22,23]. GWAS summary statistics for falls (*n* = 461,725), rheumatoid arthritis (*n* = 463,010), sleeplessness (*n* = 462,341), cataracts (*n* = 463,010), Alzheimer’s disease (*n* = 399,793), depression (*n* = 462,933), atherosclerotic heart disease (*n* = 463,010), glaucoma (*n* = 462,933), bipolar disorder (*n* = 337,159), fracture (*n* = 460,389), epilepsy (*n* = 463,010), severe stress (*n* = 337,199), anxiety disorder (*n* = 463,010), headache (*n* = 463,010) and patient death (*n* = 462,235) all from GWAS pipeline derived variables from UK Biobank. In addition, GWAS summary statistics for BMI (*n* = 236,781) and type 2 diabetes (*n* = 655,666) were generated by meta-analysis in European adults [24,25]. GWAS summary statistic for Parkinson’s disease (*n* = 482,730) came from International Parkinson’s Disease Genomics Consortium, and GWAS summary statistic for stroke (*n* = 446,696) came from a multi-ancestry genome-wide association study [26].

### 2.2. Selection of Instrumental Variables

To filter eligible genetic instrumental variables (IVs) that fulfill the three core MR assumptions, we performed a set of quality control techniques. Firstly, we chose SNPs with a strong association (*p* < 5 × 10^−7^). Secondly, to exclude SNPs in strong linkage disequilibrium (LD), we performed the clumping procedure with *R*^2^ < 0.001 and a window size = 10,000 kb with the European ancestral individuals from the 1000 Genomes Project. Thirdly we also calculated R^2^ to estimate the proportion of phenotypic variance explained and F-statistics to evaluate the strength of the instruments [27]. We selected SNPs with an F-statistic greater than 10 as the instrumental variable, indicating no weak instrumental variables bias. Detailed information on selected SNPs was summarized in Appendix A.

### 2.3. Statistical Analysis for Mendelian Randomization

The MR method is based on the following InSIDE hypothesis: Genetic variants are associated with the exposure factor; genetic variants must not be related to any confounding factors that are associated with the outcome; genetic variants must influence the outcome through exposure factors rather than through alternative way (shown in Figure 1A).

This study applied multiple complementary approaches, including the inverse variance weighted (IVW), the weighted median (WM), and the MR–Egger regression to estimate the causal effects of exposures on outcomes. The IVW method was used as the major analysis method. The IVW method yields a consistent causal estimate by combining the Wald ratios of the causal effects of each SNP, but this may also introduce ineffective IVs [27,28]. The WM estimate provides a valid estimate if at least 50% of the weight is from effective IVs [29].

At the same time, to investigate the direct effect of risk factors, we performed a multivariable MR analysis [30], which is an extension of univariate MR and can jointly detect the causal effects of multiple risk factors [31]. Multivariable MR takes into account the relationship between exposures and outcomes.

### 2.4. Mediation Analysis to Explore the Mediation Effect of Falls in the Path from Exposure to Outcome

New evidence showed that falls account for 70 percent of accidental deaths in people 75 and older, more than 90 percent of hip fractures are caused by falls [32], and that fall prevention is often included in fracture prevention recommendations for older adults. Nevertheless, since the causal relationship between fall risk factors and fracture, epilepsy, and stroke was confirmed in our analysis, a natural and direct question was whether falls modulate the effects of these risk factors on fracture and epilepsy. To address this problem, we further conducted a mediation analysis of falls as mediators to assess the mediating role of falls. Briefly, we estimated the causal effect of falls on fracture and epilepsy with IVW methods and estimated the causal effect of falls on fracture, epilepsy, and stroke by the multivariable MR analysis. Such an analysis is also referred to as network MR (shown in Figure 1B) [33].

### 2.5. Pleiotropy and Sensitivity Analysis

As a sensitivity analysis, we used the MR–Egger method, which can explore and adjust for pleiotropy [34]. However, the MR–Egger method may be inaccurate, especially when the correlation coefficients between SNP and exposure are similar or when the number of genetic instruments is small [35]. The heterogeneity estimated by Cochran’s Q test was to appraise whether any single IV was driving the results and to check for consistency of the analyses with MR assumptions. We have also made different diagnostic plots to describe the robustness of the causal estimates of the MR analyses. The scatter plots present the relationship of SNP-exposure association against SNP-outcome association, while the forest plots visualize the contribution of individual instrumental variables to the overall causal estimation. Leave-one-out analyses were used to recalculate the causal estimates from IVW by dropping out one SNP at a time to verify if the estimates were biased or driven by an outlier.

All statistical tests were two-sided and were considered to show statistical significance at a *p*-value below 0.05. Our statistical analysis was mainly conducted using the “TwoSampleMR” and “MendelianRandomization” packages for R (version 3.5.2) software. R: A language and environment for statistical computing.

## 3. Results

### 3.1. Causal Effect of Risk Factors on Falls

The numbers of the selected SNPs for MR analyses are shown in Appendix A. The MR results using three methods are shown in Table 1 and Figure 2. First, IVW analysis found that waist circumference (*P*_IVW_ < 0.001), hip circumference (*P*_IVW_
*<* 0.001), rheumatoid arthritis (*P*_IVW_ = 0.001), insomnia (*P*_IVW_ < 0.001), BMI (*P*_IVW_ = 0.003) and osteoporosis (*P*_IVW_ = 0.006) were positively associated with falls. The further weighted median method showed that all risk factors, except osteoporosis (*P*_weighted median_ = 0.353), maintained a causal relationship with falls in line with the IVW method. However, when using the MR–Egger method, we observed only one significant positive causal association between the hip circumference and falls (*P*_MR_–_Egger_ = 0.007). The MR–Egger intercepts suggest directional pleiotropy is not biasing the estimates (shown in Table 1). For the results with heterogeneity by Cochran’s Q test (shown in Appendix A), we tested them using a random effects model, which showed that there were still causal effects (*p* < 0.05). To detect outliers, we also plotted causality scatter plots and funnel plots; only the rs76895963 and rs113851554 were identified as the outliers for hip circumference on falls, and sleeplessness on falls, respectively (shown in Appendix A), and the MR results remained significant after excluding the outliers (shown in Appendix A and Appendix A). In addition, the results of the “single-SNP” and “leave-one-out” methods showed that rs6684375 significantly affected the correlation of osteoporosis with falls; and rs113851554 showed a significant effect on the correlation of sleeplessness with falls (shown in Appendix A).

### 3.2. Causality between Falls and Outcomes

In the Mendelian randomization of falls and outcomes, we chose independent SNPs significantly associated with falls as instrumental variables for the outcomes: epilepsy, fracture, headache, death, anxiety disorder, severe stress, and stroke, respectively (shown in Appendix A). 

The IVW analysis showed that an increase in falls leads to an increased risk of fracture (*p* < 0.001), epilepsy (*p* = 0.009), and stroke (*p* = 0.003), and the weighted median method showed results consistent with those shown by the IVW method (shown in Table 2 and Figure 3). The results excluded heterogeneity and horizontal pleiotropy in causality by Cochran’s Q test and MR–Egger intercept test (shown in Table 2 and Appendix A). The scatter and funnel plots showed no potential outliers that could influence the causal relationship (shown in Appendix A). The results of the “single-SNP” and “leave-one-out” methods showed that no SNP with a large effect size could bias the estimation of the causal links (shown in Appendix A).

### 3.3. Multivariable MR Analyses

Previous observational studies have shown that BMI, waist circumference, and hip circumference are associated with osteoporosis [36]. Therefore, we estimated the effects of sleeplessness, osteoporosis, hip circumference, waist circumference, and BMI on falls using multivariable MR and observed that sleeplessness, osteoporosis, hip circumference, waist circumference, and BMI had OR values of 1.139 (95% CI = 1.063–1.220, *p* < 0.001), 1.602 (95% CI = 0.964–2.662, *p* = 0.069), 1.033 (95% CI = 0.999–1.070, *p* = 0.061), 1.009 (95% CI = 0.963–1.058, *p* = 0.705) and 1.016 (95% CI = 0.982–1.052, *p* = 0.350) (shown in Figure 4 and Appendix A). After adjusting for other risk factors, sleeplessness still had a significant effect on falls.

### 3.4. Results of the Mediation Analysis

Further, we estimated the mediation effects of falls on risk factors and outcomes and constructed the causal chains from risk factors → falls → falls outcomes; we combined the instrumental variables of risk factors, falls, and outcomes in the mediation analysis. Firstly, we performed the MR analyses for the paired risk factors with outcomes. From the univariable MR analyses, we observed that increased hip circumference, sleeplessness, and osteoporosis have a positive effect on the occurrence of fracture (OR = 1.007, 95% CI = 1.002–1.012, *p* = 0.003; OR = 1.031, 95% CI = 1.009–1.052, *p* = 0.004; OR = 1.706, 95% CI = 1.195–2.437, *p* = 0.003). The effect of hip circumference and BMI was also significant in stroke (OR = 1.112, 95% CI = 1.037–1.193, *p* = 0.003; OR = 1.097, 95% CI = 1.002–1.202, *p* = 0.045). There was a causal association between waist circumference and epilepsy (OR = 1.002, 95% CI = 1.000–1.003, *p* = 0.018) (shown in Appendix A and Appendix A). For the risk factors and outcomes with causal correlations, we further conducted a mediation analysis to determine the mediating effects of falls. Finally, three mediation effects of falls were observed on the hip circumference and fracture (*p* < 0.001), waist circumference and epilepsy (*p* < 0.001), as well as sleeplessness and fracture (*p* < 0.001), which constructed three casual chains (i.e., hip circumference → falls → fracture, waist circumference → falls → epilepsy, and sleeplessness → falls → fracture) (shown in Table 3 and Figure 5).

## 4. Discussion

This study analyzed the causal effects between risk factors and falls and between falls and different outcomes utilizing MR analysis. We found that increased waist circumference, hip circumference, rheumatoid arthritis, insomnia, BMI, and osteoporosis were significantly associated with an increased risk of falls, and an increased risk of falls also led to an increased risk of fracture, epilepsy, and stroke. Our further MVMR found that sleeplessness may be an important risk factor for falls. Finally, mediation analysis showed that falls might play a mediating role in insomnia leading to fractures, waist circumference leading to epilepsy, and hip circumference leading to fractures.

Previous studies have found extensive associations between risk factors and falls based on cross-sectional analyses. For example, a cross-sectional analysis based on UK Biobank baseline data found that the patients with rheumatoid arthritis were associated with reported falls in the last year (Men: OR = 1.54, *p* < 0.001; Women: OR = 1.36, *p* < 0.001) [37]. A population-based study of chronic disease and falls in Canada identified that arthritis (OR, 95% CI = 22.9–25.9, *p* < 0.0001) and osteoporosis (OR, 95% CI = 23.2–27.7, *p* < 0.0001) were associated with falls [38]. A study of 34,163 elderly nursing home residents reported that insomnia (OR = 1.52, 95% CI = 1.38–1.66) predicted a significantly greater risk of falls, and a community-based study also reported that poor sleep was an independent risk factor of falls (OR = 1.36, 95% CI = 1.07–1.74) after adjusting for confounding factors [39,40]. There have been studies showing that BMI, an indicator of physical obesity, and waist and hip circumference, indicators of abdominal obesity, showed associations with falls. Compared to the healthy BMI group, the high BMI group showed a significant correlation with falls ≥ 1 time (OR, 95% CI = 1.02–1.10, *p* < 0.001) [41]. The obese group (BMI greater than 30 kg/m^2^) reported a higher prevalence of falls [42]. A cross-sectional study of menopausal women in Spain showed that a waist-to-hip ratio greater than 0.86 was associated with falls in menopausal women [12]. These associations from cross-section analyses can only show that certain relationships exist but cannot determine their associations due to the causal effect. Our study helps to clarify their relationship, and the results will benefit the prevention and intervention of falls in the future.

As a serious consequence of falls, epilepsy is positively correlated with falls. About 11.4% of patients with fall-induced traumatic brain injury developed post-traumatic seizures, and the odds of post-traumatic seizures were higher after a fall from a height [43]. Based on self-reported information from a behavioral risk factor detection system, a history of stroke was associated with a significantly increased risk of falls in older adults [44]. In addition, a population-based prospective study observed that an increase of 1.75 falls per year may be followed by a doubling of the incidence of hip fractures and distal forearm fractures [45]. Our MR results also support these studies.

The underlying mechanisms for the causal associations between risk factors and falls can be partially inferred but remain largely unknown. For example, physical parameters were the risk factors for falls. BMI may increase bone marrow adipogenesis, up-regulate pro-inflammatory cytokines and decrease calcium uptake, leading to falls [46]. High BMI also causes the forward movement of the whole body’s center of mass (COM), which impairs static and dynamic stability and thus affects trunk posture when standing and walking [47]. Obesity is also associated with a wide range of musculoskeletal conditions that may influence bodily movement and postural stability leading to more falls [48,49,50]. Meanwhile, waist circumference is a measure of abdominal obesity, which may adversely affect skeletal ‘quality’ (e.g., bone microarchitecture, cortical porosity, bone matrix, mineralization, collagen deposition, geometry, and bone connectivity in three dimensions) and lead to fractures, which may be more local or paracrine rather than systemic in nature [51]. For osteoporosis as a risk of falls, numerous previous studies showed that muscle mass and good posture alignment were critical for balance control in older adults, and osteoporosis patients often have muscle weakness and an increase in kyphosis, leading to vertebral fractures and poor balance control, and even falls [52]. Patients with rheumatoid arthritis may be at greater risk for falls due to altered gait, poor mobility and balance, muscle weakness, brittle bones, pain, and fatigue [53,54,55,56]. RA can also limit joint range of motion, impair gait and mobility and decrease plantar sensation [57,58]. 

The mechanisms regarding the link between sleep and falls are more complex. First, Kanda et al. reported that sleep deprivation was associated with cerebral white matter lesions, which have been demonstrated as a strong risk factor for falls in the elderly population [59,60]. Second, deprived sleep has been reported to be linked with an elevated tumor necrosis factor-alpha level. This elevated level was associated with higher reaction time, memory problems, and damaged attention, all of which were risk factors for falls [61,62]. Third, short sleep duration due to insomnia, sleep fragment, and poorer sleep quality could cause poor physical performance, which may also lead to an increased risk of falls [63,64]. Finally, people with insomnia, even primary insomnia, can feel sluggish, tired, slow, and lethargic as a result of poor sleep. Thus, sleep deprivation, or any condition that leads to sleep deprivation, may induce cognitive and psychomotor deficits that can lead to falls and fractures [65].

Fractures due to violent impacts are common in falls. For stroke and epilepsy due to falls, we considered that it might be due to Traumatic Brain Injury (TBI). Falls are the most significant factor in hospital admissions for TBI. In addition, TBI is a risk factor for a variety of neurological disorders, including epilepsy, stroke, and neurodegenerative diseases. Therefore, we can assume that brain injury caused by falls can further lead to stroke and epilepsy [66,67].

The great advantage of this study over traditional observational studies is that the causal estimates obtained by MR Avoid reverse causality and confounding bias. Our study is also the largest Mendelian randomization study on falls to date, which greatly improves the accuracy of the estimated effect. Compared with previous studies, we systematically explored the risk factors and the outcomes caused by falls and obtained three causal chains from risk factors → falls → falls outcomes. However, several limitations cannot be avoided in our study. First, although SNPs used as instrumental variables are effective in GWAS, they may increase the likelihood of false positives due to sample size limitations. The presence of weaker IVs can skew the results [68]. Second, the relatively small number of SNPs as IVs can explain only a limited causal relationship [69]. Upon combining multiple genetic variations, statistical power can be promoted effectively, and more accurate estimates can be obtained [70]. Third, our study population is mainly from Europe. However, the effects of some chronic diseases and human parameters on humans may depend on race and environment, though the results can be inferred in other populations. Finally, because the public GWAS data on falls used in our study were not further subdivided by age, sex, and fall frequency, there are some limitations in causal inference for subgroups of falls.

## 5. Conclusions

In conclusion, our study provides an unprecedented and comprehensive screening of risk factors for falls and outcomes resulting from falls. Our study increases our understanding of the risk factors for falls and the severity of falls, which will be beneficial to identify the population with a high risk of falls so as to take further preventive measures, which can effectively avoid serious injuries caused by falls.

## Figures and Tables

**Figure 1 healthcare-11-01889-f001:**
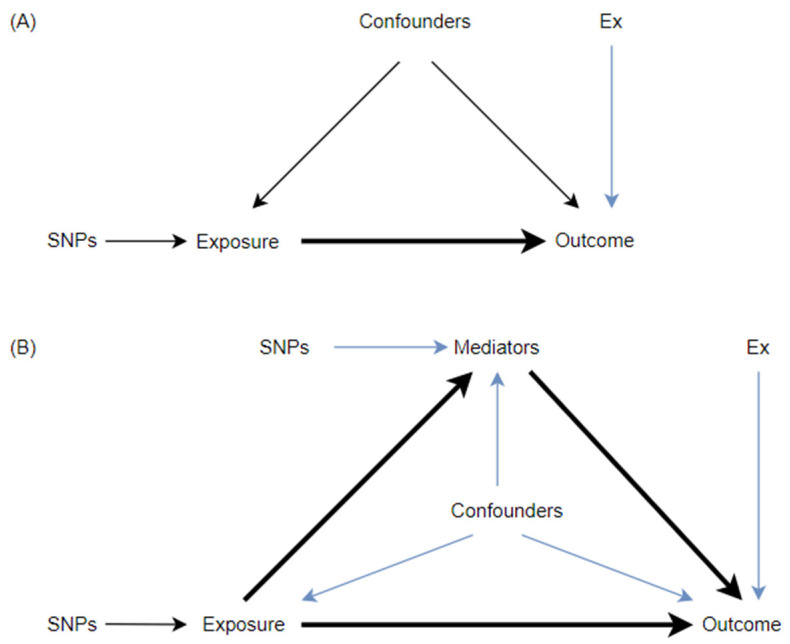
Relationship of variables in the Mendelian randomization. (**A**) between exposure and outcome; (**B**) among exposure, mediator, and outcome in the Mendelian randomization.

**Figure 2 healthcare-11-01889-f002:**
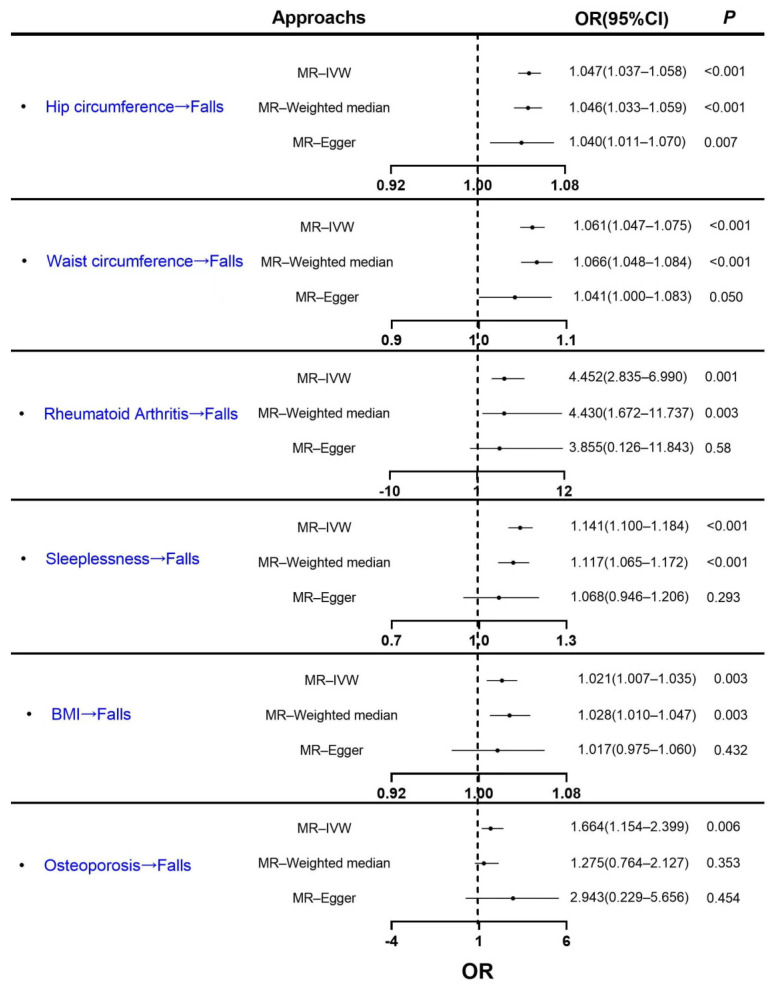
The causal effects of fall risk factors (waist circumference, rheumatoid arthritis, sleeplessness, BMI, osteoporosis, and hip circumference) on falls by univariable MR. These shown data are odds ratios and 95% confidence intervals.

**Figure 3 healthcare-11-01889-f003:**
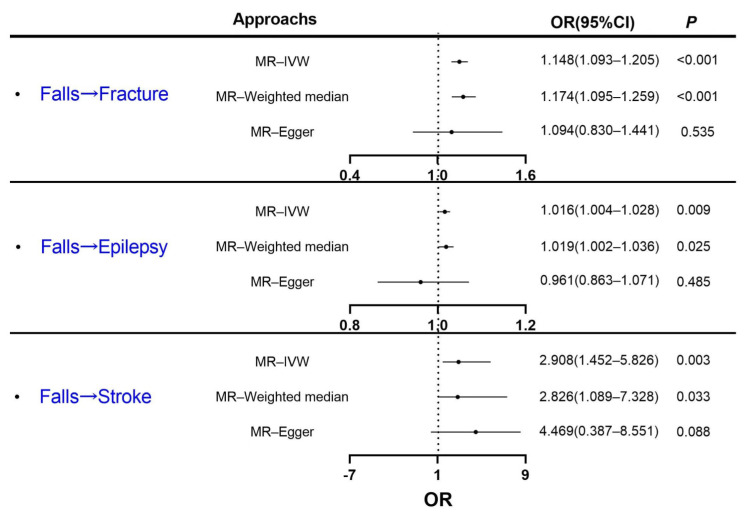
The causal effects of falls on fracture, epilepsy, and stroke using univariable MR. These shown data are odds ratios and 95% confidence intervals.

**Figure 4 healthcare-11-01889-f004:**
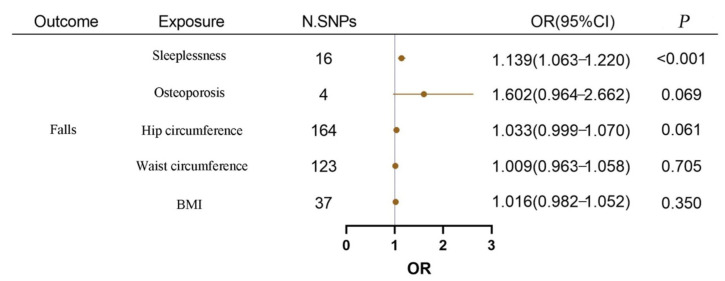
The causal effects of sleeplessness, osteoporosis, hip circumference, waist circumference, and BMI on falls were estimated using a multivariate MR–IVW method. These shown data are odds ratios and 95% confidence intervals.

**Figure 5 healthcare-11-01889-f005:**
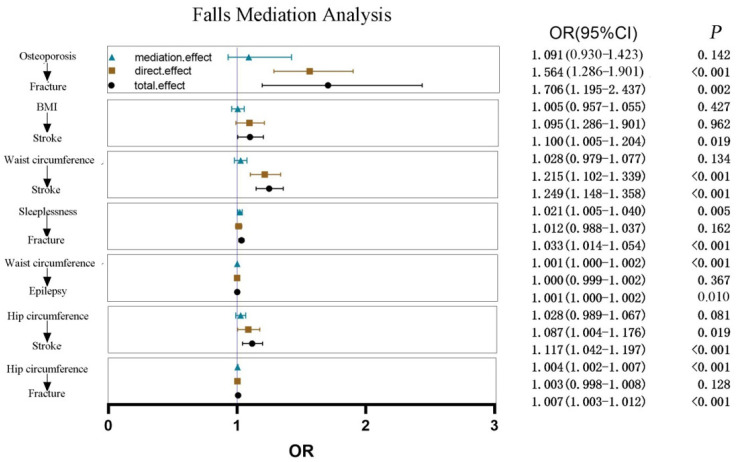
Forest plot of mediation analysis of osteoporosis on fracture, BMI on stroke, waist circumference on stroke, sleeplessness on fracture, waist circumference on epilepsy, hip circumference on stroke, and hip circumference on fracture with fall concentrations.

**Table 1 healthcare-11-01889-t001:** Causal effects of risk factors on falls estimated by using MR–Egger, MR–Weighted median, and MR–IVW.

Exposure	Outcome	Sample Size	MR–IVW		MR–Weighted Median		MR–Egger		MR–Egger Intercept	Intercept *p* Value
			OR (95%CI)	*p*	OR (95%CI)	*p*	OR (95%CI)	*p*		
Weight	Falls	336,227	1.051 (1.042–1.061)	0.579	1.041 (1.028–1.054)	0.522	1.035 (1.010–1.061)	0.468	−0.0019	0.406
Height	Falls	253,288	1.003 (0.997–1.008)	0.362	1.004 (0.997–1.012)	0.242	1.009 (0.994–1.023)	0.253	−0.0002	0.289
Sitting height	Falls	336,172	0.997 (0.989–1.005)	0.443	0.997 (0.986–1.007)	0.536	0.995 (0.977–1.013)	0.554	0.0001	0.765
Hip circumference	Falls	336,601	1.047 (1.037–1.058)	<0.001	1.046 (1.033–1.059)	<0.001	1.040 (1.011–1.070)	0.007	0.0001	0.607
Waist circumference	Falls	336,639	1.061 (1.047–1.075)	<0.001	1.066 (1.048–1.084)	<0.001	1.041 (1.000–1.083)	0.050	0.0003	0.320
Waist-hip ratio	Falls	85,978	0.833 (0.637–1.091)	0.185	0.672 (0.459–0.985)	0.042	0.663 (0.349–1.261)	0.225	0.0006	0.450
BMI	Falls	236,781	1.021 (1.007–1.035)	0.003	1.028 (1.010–1.047)	0.003	1.017 (0.975–1.060)	0.432	0.0001	0.857
Rheumatoid Arthritis	Falls	463,010	4.452 (2.835–6.990)	0.001	4.430 (1.672–11.737)	0.003	3.855 (0.126–11.843)	0.580	0.0003	0.946
Sleeplessness	Falls	462,341	1.141 (1.100–1.184)	<0.001	1.117 (1.065–1.172)	<0.001	1.068 (0.946–1.206)	0.293	0.0007	0.263
Osteoporosis	Falls	337,159	1.664 (1.154–2.399)	0.006	1.275 (0.764–2.127)	0.353	2.943 (0.229–5.656)	0.454	−0.0012	0.686
Type 2 diabetes	Falls	655,666	1.002 (0.998–1.007)	0.32	0.999 (0.994–1.006)	0.916	0.997 (0.987–1.007)	0.600	0.0004	0.283
Cataract	Falls	463,010	0.887 (0.539–1.459)	0.636	0.899 (0.511–1.585)	0.715	0.433 (0.111–1.687)	0.267	0.0018	0.305
Alzheimer’s disease	Falls	399,793	0.744 (0.519–1.065)	0.106	0.681 (0.435–1.068)	0.094	0.480 (0.106–2.184)	0.413	0.0015	0.601
Parkinson’s disease	Falls	482,730	0.997 (0.992–1.002)	0.324	0.996 (0.991–1.002)	0.224	0.991 (0.976–1.004)	0.207	0.0010	0.332
Depression	Falls	462,933	1.172 (0.784–1.752)	0.439	0.928 (0.541–1.591)	0.785	4.944 (0.706–34.645)	0.206	−0.0044	0.235
Atherosclerotic heart disease	Falls	463,010	0.949 (0.784–1.150)	0.595	0.995 (0.786–1.261)	0.968	1.338 (0.860–2.080)	0.206	−0.0012	0.103
Glaucoma	Falls	462,933	0.952 (0.642–1.404)	0.795	0.957 (0.563–1.625)	0.869	0.615 (0.186–2.032)	0.434	0.0007	0.459
Stroke	Falls	446,696	1.009 (0.998–1.019)	0.100	1.006 (0.993–1.019)	0.374	0.997 (0.941–1.057)	0.926	0.0007	0.700
Bipolar disorder	Falls	337,159	0.234 (0.042–1.317)	0.099	0.191 (0.024–1.487)	0.114	0.338 (0.000–1.981)	0.829	−0.0011	0.940

**Table 2 healthcare-11-01889-t002:** Causal effects of falls on their outcomes estimated by using MR–Egger, MR–Weighted median, and MR–IVW.

Exposure	Outcome	Sample Size	MR–IVW		MR–Weighted Median		MR–Egger		MR–Egger Intercept	Intercept *p* Value
			OR (95%CI)	*p*	OR (95%CI)	*p*	OR (95%CI)	*p*		
Falls	Fracture	460,389	1.148 (1.093–1.205)	<0.001	1.174 (1.095–1.259)	<0.001	1.094 (0.830–1.441)	0.535	0.0004	0.733
Falls	Epilepsy	463,010	1.016 (1.004–1.028)	0.009	1.019 (1.002–1.036)	0.025	0.961 (0.863–1.071)	0.485	0.0004	0.329
Falls	Stroke	446,696	2.908 (1.452–5.826)	0.003	2.826 (1.089–7.328)	0.033	4.469 (0.387–8.551)	0.088	−0.0205	0.203
Falls	Severe stress	337,199	1.000 (0.995–1.005)	0.971	0.998 (0.991–1.004)	0.478	1.005 (0.975–1.035)	0.749	−3.70 × 10^−5^	0.750
Falls	Anxiety disorder	463,010	0.999 (0.985–1.012)	0.844	0.998 (0.982–1.015)	0.806	1.001 (0.736–1.362)	0.993	−1.90 × 10^−5^	0.986
Falls	Headache	463,010	1.006(0.990–1.023)	0.430	0.999 (0.978–1.022)	0.962	0.932 (0.817–1.062)	0.308	0.0006	0.265
Falls	Patient death	462,235	1.005 (0.995–1.016)	0.338	1.005 (0.992–1.019)	0.452	1.046 (0.834–1.311)	0.709	−0.0003	0.741

**Table 3 healthcare-11-01889-t003:** Mediation effects of falls between the risk factors and outcomes.

Exposure	Mediator	Outcome	Total Effect OR (95% CI)	Direct Effect OR (95% CI)	Mediation Effect OR (95% CI)	Mediated *p*-Values	Proportion
Osteoporosis	Falls	Fracture	1.706 (1.195–2.437)	1.564 (1.286–1.901)	1.091 (0.930–1.423)	0.142	-
BMI	Falls	Stroke	1.100 (1.005–1.204)	1.095 (1.286–1.901)	1.005 (0.957–1.055)	0.472	-
Waist circumference	Falls	Stroke	1.249 (1.148–1.358)	1.215 (1.102–1.339)	1.028 (0.979–1.077)	0.134	-
Sleeplessness	Falls	Fracture	1.033 (1.014–1.054)	1.012 (0.988–1.037)	1.021 (1.005–1.040)	0.005	63.64%
Waist circumference	Falls	Epilepsy	1.001 (1.000–1.002)	1.000 (0.999–1.002)	1.001 (1.000–1.002)	<0.001	100%
Hip circumference	Falls	Stroke	1.117 (1.042–1.197)	1.087 (1.004–1.176)	1.028 (0.989–1.067)	0.081	-
Hip circumference	Falls	Fracture	1.007 (1.003–1.012)	1.003 (0.998–1.008)	1.004 (1.002–1.007)	<0.001	57.14%

## Data Availability

Data are available in a public, open-access repository. Data URLs: https://gwas.mrcieu.ac.uk (accessed on 20 June 2022); https://www.ebi.ac.uk/gwas/down-loads/summary-statistics (accessed on 20 June 2022).

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
