# Peer review of "The Casual Association Inference for the Chain of Falls Risk Factors-Falls-Falls Outcomes: A Mendelian Randomization Study"

_healthcare, 2023, doi:10.3390/healthcare11131889_

Round 1
Reviewer 1 Report
Thank you very much for your study, I found it very interesting. I would like to leave you a tip to change your article.
In the last paragraph of the introduction, the results of your study appear. In this section, a justification of the study and your hypothesis should appear, but no results should appear.
Author Response
Point 1: In the last paragraph of the introduction, the results of your study appear. In this section, a justification of the study and your hypothesis should appear, but no results should appear.
Response 1: Thank you for your suggestion. In the last paragraph of the introduction on page two, we have changed it to "To infer falls causal relationships, this study performed systemic causal inferences for two pairs (risk factors and falls, falls and outcomes), and further constructed the causal chains from risk factors → falls → falls outcomes by using the publicly available GWAS summary data.“

Reviewer 2 Report
The authors described the causal association of chain of falls risk factors-falls-falls outcomes. However, the risk factors of falls has been reported widely reported not only based on cross-sectional study, but also longitudinal study. Here are some comments to improve the manuscript.
1. Please briefly describe your study population or what are the data that being used for the present analysis in abstract.
2. In the introduction part, the authors discussed about the falls among older adults. May I know if the data included in the analysis was data extracted from older adults only or include subjects in all age group? Such data is missing in the methodology part. The risk factors of falls for different age group might be different.
3. Previous studies showed that the risk factors for single fall and repeated falls might be different. Is falls frequency is being taken into considerations in this analysis?
4. For the results part, please follow the standard reporting format instead of putting the figures and tables under one subsection.
5. In the first paragraph of discussion, the authors mentioned that "Finally, mediation analysis showed that falls may play a mediating role in insomnia leading to falls". Please double check if this statement is correct. Is it falls may play a mediating role in insomnia leading to fracture?
6. What is the sample size for this study? Please state it in the writing.
7. The authors should better describe the novelty of the study as most of the risk factors and adverse impact of falls have been reported elsewhere.
Author Response
Point 1: Please briefly describe your study population or what are the data that being used for the present analysis in abstract.
Response 1: The data we analyzed were mainly from the UK Biobank. UK Biobank has collected an unprecedented amount of biological and medical data on half a million people, aged between 40 and 69 years old and living in the UK, as part of a large-scale prospective study. And in the abstract, we added "Our study data are mainly from White European individuals (40-69 years) from the UK Biobank. ”
Point 2: In the introduction part, the authors discussed about the falls among older adults. May I know if the data included in the analysis was data extracted from older adults only or include subjects in all age group? Such data is missing in the methodology part. The risk factors of falls for different age group might be different.
Response 2: The GWAS summary of falls in our study was obtained from the UK Biobank, which included participants aged 40 to 69 years. Therefore, our study discussed older people and didn’t consider the effect of different age groups. This is also illustrated by our limitation in the last paragraph of the discussion on page 11: "Finally, because the public GWAS data on falls used in our study were not further subdivided by age, sex, and fall frequency, there are some limitations in causal inference for subgroups of falls.”
Point 3: Previous studies showed that the risk factors for single fall and repeated falls might be different. Is falls frequency is being taken into considerations in this analysis?
Response 3: A fall case in the UK Biobanks was defined as a participant who gave a positive answer to the following question: " In the last year have you had any falls?". Previous GWAS studies of falls only divided the population into falls and non-falls, so our study did not consider the frequency of falls. And this is also illustrated by our limitation in the last paragraph of the discussion on page 11: "Finally, because the public GWAS data on falls used in our study were not further subdivided by age, sex, and fall frequency, there are some limitations in causal inference for subgroups of falls.”
.
Point 4: For the results part, please follow the standard reporting format instead of putting the figures and tables under one subsection.
Response 4: Thank you for your suggestion, and I've revised it.
Point 5: In the first paragraph of discussion, the authors mentioned that "Finally, mediation analysis showed that falls may play a mediating role in insomnia leading to falls". Please double check if this statement is correct. Is it falls may play a mediating role in insomnia leading to fracture?
Response 5: Thank you for your advice. And We have revised it to " Finally, mediation analysis showed that falls may play a mediating role in insomnia leading to fractures ".
Point 6: What is the sample size for this study? Please state it in the writing.
Response 6: Thank you for your advice. We have explained the sample size in the table, and we have stated it in our Section 2.1 on page 2. ”GWAS summary statistics of waist circumference (N=336639), hip circumference (N=336601), Waist-hip ratio (N=85978), Sitting height (N=336172), and osteoporosis (N=337159) are available online. Data on height (N=253288) and weight (N=336227) were obtained from GIANT consortium data files. GWAS summary statistics for falls (N=461725), rheumatoid arthritis (N=463010), sleeplessness (N=462341), cataract (N=463010), Alzheimer's disease (N=399793), depression (N=462933), atherosclerotic heart disease (N=463010), glaucoma (N=462933), bipolar disorder (N=337159), fracture (N=460389), epilepsy (N=463010), severe stress (N=337199), Anxiety disorder (N=463010), headache (N=463010) and patient death (N=462235) all from GWAS pipeline derived variables from UK Biobank. Besides GWAS summary statistics for BMI (N=236781) and type 2 diabetes (N=655666) were generated by meta-analysis in European adults. GWAS summary statistic for Parkinson's disease (N=482730) came from International Parkinson's Disease Genomics Consortium. And GWAS summary statistic for stroke (N=446696) came from a multi ancestry genome-wide association study.”
Point 7: The authors should better describe the novelty of the study as most of the risk factors and adverse impact of falls have been reported elsewhere.
Response 7: Thank you for your advice. In the discussion section of the last paragraph on page 10, we illustrate “The great advantage of this study over traditional observational studies is that the causal estimates obtained by MR Avoid reverse causality and confounding bias. Our study is also the largest Mendelian randomization study on falls to date, which greatly improves the accuracy of the estimated effect. Compared with previous studies, we systematically explored the risk factors and the outcomes caused by falls, and obtained three causal chains from risk factors → falls → falls outcomes.”

Reviewer 3 Report
The onset of a fall in an elderly person is multifactorial. It depends on predisposing health factors, but also on the patient's behavior and the environment.
Health factors can be predisposing, constituting the risk terrain or genetic risk. They can also be factors of severity, having an impact on the direct complications of a fall. This study looks at these two factors of predisposition and severity of complications.
The title does not reflect the content of the article. It should include the genetic context.
l 86 phesant Explain acronym (PHEnome Scan ANalysis Tool)
Results
The identification of factors causing falls is clear, but can we assign them a risk coefficient when they are isolated or associated? (like falls-outcomes)
Discussion
l 266-7 Can you determine a genetic score that predicts the risk of falls at an early stage?
Conclusion
Ok for screening, but "useful recommendations" are not clear. What is the clinical benefit of the genetic score?
Author Response
Point 1: The onset of a fall in an elderly person is multifactorial. It depends on predisposing health factors, but also on the patient's behavior and the environment.
Health factors can be predisposing, constituting the risk terrain or genetic risk. They can also be factors of severity, having an impact on the direct complications of a fall. This study looks at these two factors of predisposition and severity of complications.
The title does not reflect the content of the article. It should include the genetic context.
Response 1: Since the study used the Mendelian randomization to infer the causal effects by using the SNPs as genetic instrumental variables, we modified the title as follows: The casual association inference for the chain of falls risk factors-falls-falls outcomes: a Mendelian randomization study
Point 2: Results: The identification of factors causing falls is clear, but can we assign them a risk coefficient or associated? (like falls-outcomes)
Response 2: Thank you for your advice. In our study, we performed univariate and multivariate Mendelian randomization. We used univariate Mendelian randomization to account for risk factors when they are isolated. And we observed that hip circumference, waist circumference, rheumatoid arthritis, sleeplessness, BMI, and osteoporosis had OR values of 1.047, 1.601, 4.452, 1.141, 1.021, and 1.664. And when we assign them associated, since these risk factors may influence each other, we performed a multivariable Mendelian randomization study on these risk factors. After excluding their mutual influence, we found that sleeplessness may be the most influential factor. (OR=1.139)
Point 3: Discussion: Can you determine a genetic score that predicts the risk of falls at an early stage?
Response 3: Thank you for your suggestion. Our study was a causal exploration of falls using Mendelian randomization. Mendelian randomization is defined as "instrumental variable analysis using genetic variation". In Mendelian randomization, genetic variants are used as instrumental variables to assess the causal effect of exposure on outcomes. Calculating genetic scores for falls requires GWAS summary data for falls and genotype data for the population, which requires additional analyses. Therefore, Calculating genetic scores is another study. And we didn’t calculate a genetic score to predict the risk of falls.
Point 4: Conclusion: Ok for screening, but "useful recommendations" are not clear. What is the clinical benefit of the genetic score?
Response 4: Thank you for your question. Our study provides an unprecedented and comprehensive screening of risk factors for falls and outcomes resulting from falls and constructs three causal chains (hip circumference → falls → fracture, waist circumference → falls → epilepsy, insomnia → falls → fracture), and the interference of confounding factors was excluded. Therefore, we can conclude that early intervention and prevention for people with high hip circumference, high waist circumference, and insomnia can effectively prevent falls and further injuries caused by falls. So we changed that in the conclusion on page 11 "Our study increase our understanding on the risk factors for falls and the severity of falls, which will be beneficial to identify the population with a high risk of falls so as to take further preventive measures, which can effectively avoid serious injuries caused by falls."

Round 2
Reviewer 2 Report
The authors have addressed most of my comments. Although the study is subjected to various limitations, findings from the Mendelian randomization approach in this study provide some interesting data on falls.